# Reproducibility Report: Towards Interpreting BERT for Reading Comprehension Based QA

## Reproducibility Summary

In the paper, the authors attempt to understand BERT's exemplary performance for RCQA tasks by defining each Self-Attention Layer's role using Integrated Gradients for SQuAD v1.1 and DuoRC SelfRC datasets. After this, they follow through with experiments and analysis to infer how each layer works to predict the answer, based on the context and question.

**Scope of Reproducibility**

Ramnath et al. suggest that the initial layers focus on query-passage interaction, while the later layers focus more on contextual understanding and enhancing answer prediction. In our reproducibility plan, we aim to validate this claim and other related claims by completely replicating the authors' experiments to analyze BERT layers to understand their RCQA-specific role and their behavior on potentially confusing Quantifier Questions.

**Methodology**

Since this paper's official code is not available, we prepare our scripts and modules for processing the data and re-implement the approach as described in the paper. We refer to the original research paper to cross-check our results with their reporting. We use Google Colab's free GPU for 35-40 hours for fine-tuning the model and calculating the Integrated Gradients. The rest of the experiments can be performed on a CPU within 10-15 hours.

**Results**

Our reproduced results for all experiments support the central claim made in the paper. All of our statistics and plots agree with those in the original paper within a good margin. We have also analyzed some results beyond the paper and find that the scope of the original paper is transferable and generalizable.

**What was easy**

Using HuggingFace Transformers and Datasets for the SQuAD v1.1 was easy as we could adapt the authors' ideas to our code experiments and verify their central claim without much effort. There are also libraries readily available for Jensen-Shannon Divergence and t-SNE and could be used easily.

**What was difficult**

Re-implementing the paper was more difficult than we expected as there were ambiguities and conflicts in our approaches for Integrated Gradients calculation, as well as DuoRC preprocessing and postprocessing. There were differences in our methods of implementation, and multiple iterations had to be performed to decide upon the case to be used, which took up a lot of computational power unnecessarily.

**Communication with original authors**

We had frequent interaction with the first author via email for clarification and discussion.

Submitted to ML Reproducibility Challenge 2020. Do not distribute.

# 1   Introduction

Previous works on interpreting BERT [1] have discussed its syntactic/semantic roles on simpler Natural Language tasks like sentiment classification, syntactic/semantic tags prediction [2–4], etc. The Reading Comprehension based Question Answering (RCQA) task involves marking an answer span in a passage, given a question. Pre-BERT systems [5–7] for question-answering tasks used pre-defined layer-wise roles. An analysis of BERT for complex tasks like RCQA is challenging due to the lack of such layer roles and its large number of parameters. Ramnath et al. [8] attempt to define these layer-roles for BERT using Integrated Gradients (IG) on the RCQA task [9] for SQuAD v1.1 [10] and DuoRC SelfRC [11] datasets. Following this, they perform analysis across all Self-Attention Layers to understand how the model predicts the answers accurately. In doing so, they provide a mechanism for interpreting the BERT layers and their roles, which can be extended to other complex tasks like machine translation, cloze-style question answering, etc.

As a part of the ML Reproducibility Challenge 2020, we replicate the experiments presented in the paper from scratch and analyze if their observations and claims hold true for our implementation.

# 2   Scope of Reproducibility

The authors address the lack of pre-defined layer roles in BERT by defining the roles of each layer using IG to get word importance scores across individual layers. Then, they use analysis techniques such as Jensen-Shannon Divergence (JSD), t-Distributed Stochastic Neighbor Embedding (t-SNE) plots, Part-of-Speech (PoS) tagging, etc., to understand how each layer contributes towards predicting the correct answer. **The central claim of the paper is that the initial layers focus on query-passage interaction, while the later layers focus on contextual understanding and answer prediction**. Following are the claims that can be extracted from the paper and that we validate through our experiments:

1. The top-K important tokens show more divergence across layers than the rest of the tokens. We explore this by analyzing JSD Heatmaps in §4.1.2 and 4.2.2.

2. The importance scores of Contextual Words[1] and Answer Words increase from initial to final layers, while the importance scores of Query Words decrease. This claim is verified in §4.1.3 using semantic statistics.

3. The initial layers find Query Words[2] more important, while the final layers focus on the Answer Words. This claim is verified using visualization in §4.1.4.

4. The initial layers represent similar words together. As the layers progress, the Query Words and Answer Words come closer to each other. Eventually, all Query Words are separated from Answer and Contextual Spans[3]. We plot t-SNE representation in §4.1.5 to validate this claim.

5. Numerical Words stay close to each other in representation throughout the layers. The t-SNE plot in §4.1.5 checks this claim as well.

6. For Quantifier Questions, the importance of Numerical Words increases from the initial to the final layers, meaning that the confusing words are more important to the model towards the end. §4.1.6 discusses the relevant scores and statistics.

7. BERT has a higher confidence score on Quantifier Questions with more than one numerical entity in passage vs. Non-Quantifier Questions. §4.1.6 discusses the relevant scores.

# 3   Methodology

The authors use the official BERT fine-tuning script for SQuAD[4]. Since there is no official code for the paper yet, we implement the authors' approaches from scratch in PyTorch. We use HuggingFace's (HF) Datasets [12] and Transformers [13] for fine-tuning BERT, and Captum for Integrated Gradients.

## 3.1   Fine-tuning and Integrated Gradients

**Fine-tuning** - We use HF's BertForQuestionAnswering model and load the pre-trained checkpoint - *bert-base-uncased*. The BERT model has 12 layers, 768 hidden units per token, 12 attention heads per layer, and 110M parameters. It is pre-trained on lower-case English text gathered from Books Corpus and English Wikipedia.

---

[1]Contextual Words are words close to the Answer Words within a window.

[2]Query Words are those question words that are present in the passage.

[3]Contextual Spans are those words which appear in the same sentence as the Answer Words.

[4]Official Fine-tuning Script: https://github.com/google-research/bert/blob/master/run_squad.py

**Attributions** - We use Captum's implementation of Integrated Gradients. IG is based on the fine-tuned BERT model. **Note that our $(n + 1)^{th}$ layer corresponds to the $n^{th}$ layer of the authors' notation as we take the Embedding Layer to be Layer 0. We perform experiments on all Self-Attention Layers and the Embedding Layer**.

## 3.2 Datasets

### 3.2.1 SQuAD v1.1 (SQuAD)

We use the SQuAD v1.1[5] dataset available on HF's Datasets[6] library for its simplicity. The train/dev splits contain 87599/10570 question-answer pairs for various passages, with start position and answer text specifying the answer. Each example breaks into tokenized features[7] using a max-overlap stride of 128, and a max sequence length of 384. We do not use a max query length of 64, unlike the official fine-tuning script. We use HF Transformers' BertTokenizerFast with the pre-trained checkpoint - *bert-base-uncased*. In cases where the ground truth is absent in the context due to splitting, we mark '[CLS]' as the answer. Our final train/development sets have 88524/10784 features.

### 3.2.2 DuoRC SelfRC (DuoRC)

We use the original DuoRC SelfRC[8] train/dev splits with 60721/12961 question-answer pairs based on 4800/984 different movie plots. Additionally, a question can have multiple answers - which may or may not exist in the plot. No start positions are provided for the answers. Training BERT on a Question Answering task requires the start and end positions of the answer. Hence, we first convert the DuoRC dataset to SQuAD format. We find the answers using exact matching in the plot. We consider the following four cases for train and dev sets:

1. **No Answer exists** - We keep the example with an empty answer. There are 627/116 questions in train/dev.

2. **Single Answer** - We include single answers found in the plot using first matching index.

3. **Multiple Answers** - We take the first answer found and store first index in the train set. In dev set, we store all the answers and corresponding first indices.

4. **Answer exists, but not found in plot** - We drop such examples in train, but keep them with empty answers in dev. There are 26596/5768 such examples.

The exact details for processing DuoRC are not provided in the paper, but we assume that this process is similar to choosing a sample for fine-tuning and prediction. The authors' claims should hold true, irrespective of the sample chosen. After this first step of preprocessing, we get 34166/12961 examples in the train/dev sets, which are then processed similarly as SQuAD to get 118676/44831 tokenized features.

## 3.3 Hyperparameters

### 3.3.1 Fine-tuning

We use hyperparameters similar to the official BERT script[4] while fine-tuning our model. The train/eval batch sizes are chosen to be 6/8 after discussing with the authors. AdamW [14] optimizer is used with a learning rate of $3 \times 10^{-5}$, a weight decay of 0.01 and an epsilon of $1 \times 10^{-6}$. The training is done for 2 epochs. A polynomial learning rate scheduler with 10% of total training steps as warmup steps is used. The other default hyperparameters in HF Transformer's TrainingArguments are not changed. We did not perform any hyperparameter search because the focus of the paper is not to improve the performance of BERT on the tasks, but to analyze its layers after training.

### 3.3.2 Integrated Gradients

Although the calculation of attributions using IG is performed after training, the choices made can significantly affect the importance distributions based on these attributions. Hence, we provide a brief description of the same.

We calculate attributions on the softmax outputs of start and end logits from the BERT model. The target positions chosen are those where start and end logits have the maximum value. Based on our discussion with the authors, only those features which give the best answer for an example during the predictions are chosen. Reimann Right numerical approximation is used for calculating the integral value. The number of steps chosen is 25, and an internal batch size of 4 is used. We perform IG on only 1000 examples from the dev sets due to computational restrictions.

---

[5]SQuAD v1.1 dataset : `https://github.com/rajpurkar/SQuAD-explorer/tree/master/dataset`

[6]HuggingFace's SQuAD dataset: `https://huggingface.co/datasets/squad#dataset-description`

[7]We use "features" to denote multiple question-context pairs per example due to the max sequence length.

[8]DuoRC SelfRC dataset: `https://github.com/duorc/duorc/tree/master/dataset`.

### 3.4 Experimental Setup and Code

#### 3.4.1 Fine-tuning BERT

We use HF Transformer's Trainer to fine-tune BERT on the tokenized features for training and validation with the hyperparameters mentioned in §3.3.1. Post-training, the predictions for each feature are processed. We choose the best valid feature per example based on the best score (start + end logit) for the top 20 start and top 20 end logits. We discard spans with length above 30 tokens. We store the respective input token IDs, attention masks, predicted text, ground start/end positions, predicted start/end positions for the best feature for all examples in a JSON file. We use SQuAD v1.1[9] and SQuAD v2[10] evaluation scripts for SQuAD and DuoRC, respectively. This is done as DuoRC predictions/ground truth may contain no answers. We consider exact match (EM) and $F_1$ scores from these evaluations.

#### 3.4.2 Integrated Gradients

Taking the best features from predictions in the JSON file, we use respective input token IDs and attention masks to calculate IG. This eliminates the possibility of having multiple features per example, and improves our sample.

We create a method to find the start and end logits given the layer index and the corresponding hidden states. We calculate start and end attributions using Captum's IntegratedGradients on 1000 randomly chosen examples. We add start and end attributions and take a Euclidean norm for each token in the sequence, which is then normalized to get an importance distribution for that sequence. This is repeated for all 13 layers, including the Embedding Layer. The token-wise importance scores are stored for each layer and sample. Note that this process is similar to the algorithm described in the paper, except that they do not calculate attributions on Embedding Layer outputs.

We change the token-wise distributions to word-wise distributions by ignoring the special tokens - *[CLS]*, *[SEP]*, *[PAD]* - and adding importance scores for multiple tokens per word. The concatenation of the tokens and the addition of importance scores is based on the fact that subsequent tokens for a word start with ##. The offset mapping for each token is used to get the exact word in the passage/question. The combined scores are re-normalized to get a word-wise importance distribution. Along with this, the word-wise categories - answer, question, context - are stored for each sample based on the predicted answer spans.

#### 3.4.3 Jensen-Shannon Divergence

For JSD heatmaps, we use pair-wise JSD of all 13 layers (Embedding Layer + Self-Attention Layers). This gives us a $13 \times 13$ heatmap for each example, which is then averaged over the 1000 examples we chose during IG calculation. This helps us understand how the layer outputs are different from each other in terms of their attributions. We use the same library as the authors - *dit* [15] - to calculate JSD. Index 0 in our heatmaps represents the Embedding Layer.

The authors create two heatmaps for each of the datasets - one with top-K token importance scores retained and the rest zeroed out, and the other with top-K token importance scores zeroed out. They chose K=2 for their experiments. We create similar heatmaps based on the token-wise importance scores generated in §3.4.2 using Seaborn [16] and vary the K values in - 2,5,10.

#### 3.4.4 QA Functionality Tables

We calculate the percentage of predicted Answer Words, Query Words, and Contextual Words (within window size=5 of the Answer Spans) in the top-5 important words for the 1000 examples we chose for IG. We represent the average values in Tables 1 and 2. The Query Words are selected using lower-case exact matching in the passage. Only words in the passage are considered for the statistical analysis.

#### 3.4.5 t-SNE Representation

We plot t-SNE representations for tokens across multiple layers based on the Query Tokens, the predicted Answer Tokens, and the Contextual Spans. All the tokens from the sentence containing the predicted Answer Tokens (between two periods (.)) are chosen as the Supporting/Contextual Spans. *[PAD]* tokens are dropped, and only the context tokens are considered for plotting. The categories finally used are - *query words*, *answer spans*, *contextual words*, *[CLS]/[SEP]* and *background*. We use sklearn's t-SNE [17] with PCA initialization and 1000 iterations to represent the hidden states in 2 dimensions and plot them using Matplotlib [18].

---

[9]Link to the HuggingFace metrics - SQuAD v1.1: https://huggingface.co/metrics/squad
[10]Link to the HuggingFace metrics - SQuAD v2: https://huggingface.co/metrics/squad_v2

### 3.4.6 Quantifier Questions

We search Quantifier Questions using 'how man' and 'how much'. We use 'man' instead of 'many' because of typographical error in some questions. There are 799 such examples in SQuAD and 310 in DuoRC dev-splits. We use IG on our predicted features for these examples to get the importance distributions. Then, using the word-wise importance scores, we find out the percentage of numerical words which are tagged as "Cardinal"(CD) by NLTK's PoS Tagger [19]. Additionally, we also include phrases like "thousands", "hundreds" and "two thousand and three" using the word2number library. We calculate the percentages of Numerical Words in top-5 words out of all Numerical Words in the passage and average the values. For EM calculations on Quantifier Questions, we use the respective evaluation scripts. For calculation of the confidence, we take the maximum of sum of softmax start and end scores. Then, we average these values across the samples to get the average confidence per category.

### 3.5 Computational Requirements

We use the free NVIDIA K80/T4 GPU provided by Google Colab for training the BERT model and calculating Integrated Gradients. All other experiments are performed on an Intel i5-6200U quad-core CPU. For each dataset - fine-tuning the BERT model takes ∼5-7 hours on the GPU; prediction takes ∼10-20 GPU minutes; and processing of predictions takes ∼4-5 CPU hours; Integrated Gradients step takes ∼5-6 GPU hours per 1000 examples; JSD Heatmap generation takes ∼2-3 CPU hours for 1000 examples; while the tables are generated in negligible CPU time. For Integrated Gradients on Quantifier Question on dev splits, SQuAD takes ∼4 GPU hours, and DuoRC takes ∼1.5 GPU hours. The calculation of confidence on Quantifier/Non-Quantifier Questions takes ∼10-20 GPU minutes per dataset.

## 4 Results

### 4.1 Results reproducing original paper

#### 4.1.1 Fine-tuning

The authors achieved $F_1$ scores of 88.73 and 54.80 on SQuAD and DuoRC dev-splits. Our fine-tuned BERT model achieves 88.51 and 50.73 on our tokenized dev-splits of SQuAD and DuoRC. The performance can depend on the weight initialization of the classifier layer, and differences in the data preprocessing (§3.2). Additionally, training is a stochastic process, hence the weights learned will vary. Therefore, some variation in the performance is expected. For SQuAD, we observe a minor change of 0.2 $F_1$ score. Additionally, for DuoRC, our way of evaluation (§3.4.1) may be different from the authors, which could be the reason behind the 4 point drop in $F_1$ score.

#### 4.1.2 Jensen-Shannon Divergence Heatmaps

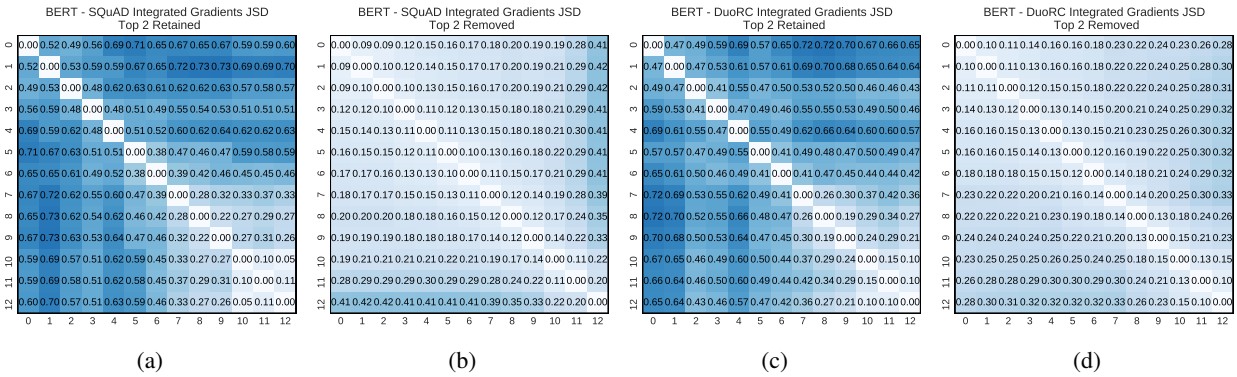

Figure 1: Jensen-Shannon Divergence Heatmaps for K=2

We plot heatmaps for the JSD values, with top-2 scores in the token-wise distribution, retained and removed, for both SQuAD and DuoRC (Figure 1). We see a similar pattern on the heatmap as the paper - the top-2 retained JSD heatmaps have a higher range for both SQuAD (0.05-0.73), and DuoRC (0.10-0.70) across the Self-Attention layers. The top-2 removed JSD heatmaps have lower ranges for SQuAD (0.10-0.42) and DuoRC (0.10-0.32). The ranges are slightly different from the paper which can be attributed to sampling. Regardless, this difference shows that the top-2 words which the layers focus on are more *different* than the rest of the words across the layers. Hence, top-K words should be chosen for analysis. Claim 1 is, thus, verified by our experiments. We repeat this analysis for K=5/10 in §4.2.2.

### 4.1.3 QA Functionality

We calculate the semantic statistics of the top-5 words for SQuAD and DuoRC as described in §3.4.4. Table 1 for 1000 examples of SQuAD follows the expected trend for Answer Words (37.58%(L1) - 42.94%(L12)), Contextual Words (33.04%(L1) - 34.02%(L12)), as well as Query Words (22.20%(L1) - 10.42%(L12)). The reasons behind slightly different percentages could be smaller sampling size and differences in counting Query and Contextual Words. For DuoRC, Table 2 shows the average statistics for the 1000 examples. While the Answer Words (11.70%(L1) - 12.94%(L12)) and Query Words(19.20%(L1) - 8.68%(L12)) follow the expected trend, the percentage of Contextual Words remains between 11-12% for the BERT Self-Attention Layers. In the paper, Contextual Words do not follow an increasing trend for DuoRC and vary between 15-33%. For DuoRC, our percentage of Answer Words is low (11-13%) compared to the paper (33-44%). In addition to the factors stated for SQuAD, DuoRC results can also deviate because of possible differences in how predictions are processed, as mentioned in §3.4.1. Hence, claim 2 holds true.

| Layer Name | Answer Words% | Contextual Words% | Q-Words% |
|---|---|---|---|
| Embedding | 38.10 | 32.96 | 22.46 |
| Layer 1 | 37.58 | 33.04 | 22.20 |
| Layer 2 | 37.10 | 33.58 | 24.08 |
| Layer 3 | 41.00 | 33.10 | 19.62 |
| Layer 4 | 40.42 | 36.40 | 16.34 |
| Layer 5 | 40.82 | 34.68 | 18.58 |
| Layer 6 | 40.74 | 36.46 | 15.62 |
| Layer 7 | 40.06 | 35.76 | 14.12 |
| Layer 8 | 41.90 | 34.94 | 11.38 |
| Layer 9 | 41.18 | 36.12 | 11.66 |
| Layer 10 | 43.36 | 35.40 | 9.74 |
| Layer 11 | 42.52 | 32.14 | 10.30 |
| Layer 12 | 42.94 | 34.02 | 10.42 |

Table 1: Semantic statistics of top-5 words - SQuAD

| Layer Name | Answer Words% | Contextual Words% | Q-Words% |
|---|---|---|---|
| Embedding | 11.78 | 9.36 | 24.00 |
| Layer 1 | 11.70 | 12.00 | 19.20 |
| Layer 2 | 12.60 | 11.84 | 17.54 |
| Layer 3 | 13.36 | 11.96 | 16.18 |
| Layer 4 | 13.16 | 12.64 | 20.30 |
| Layer 5 | 12.68 | 11.24 | 22.02 |
| Layer 6 | 12.96 | 11.72 | 15.72 |
| Layer 7 | 12.68 | 11.90 | 12.86 |
| Layer 8 | 13.36 | 12.22 | 8.24 |
| Layer 9 | 12.66 | 12.78 | 5.50 |
| Layer 10 | 12.90 | 11.12 | 6.74 |
| Layer 11 | 13.06 | 11.86 | 7.52 |
| Layer 12 | 12.94 | 11.78 | 8.68 |

Table 2: Semantic statistics of top-5 words - DuoRC

### 4.1.4 Visualization

Claim 3 says that the focus is more on query-passage interaction in the initial layers, while in the final layers more importance is given to the answer and contextual spans. From the visualized example in Figure 2, we can see that the Query Words - (*percentage/%*, *increase/increased*, *agriculture*) - are given more importance in Embedding Layer (L0), Layers 1, 2, and 3. While the Answer Words - *17* - and Contextual Words - *%* - receive more importance in the later layers. This observation shows that the attributions shifts to Contextual and Answer Words in later layers. Since this example is in agreement with the example shown in the original paper, we consider claim 3 validated.

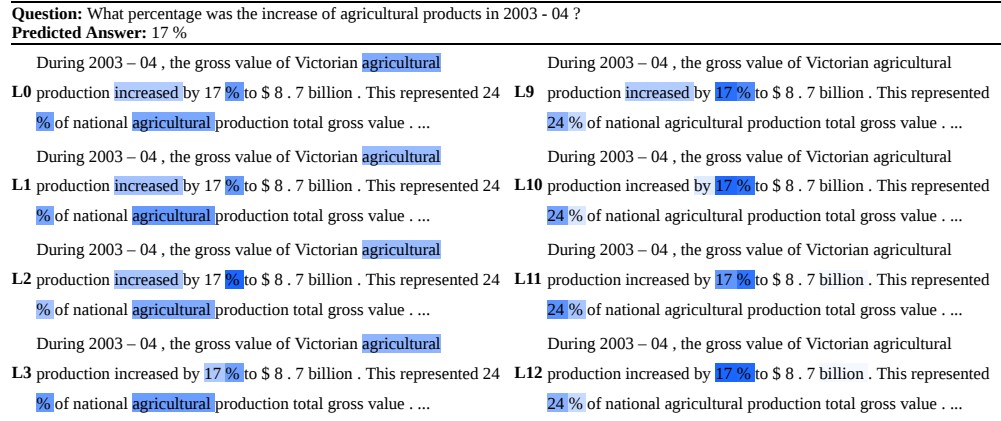

Figure 2: Qualitative Visualization of top-5 words - SQuAD

### 4.1.5 t-SNE Representation

We plot t-SNE representations in Figure 3 which verify that Claims 4 and 5 hold true. We observe that in the initial layers similar words like *california, francisco, santa, clara* are close to each other. In the later layers, the Answer Words and Query Words separate out. The Layer 12 plot shows that the model has successfully recognised and separated

Answer and Query Words. Also, BERT representations of confusing words are closer to each other in the later layers. On careful observation, we see that Numerical Words like *50*, *2015* and *50th* are close to the representation of the Answer Word *2016*. This means that BERT tends to focus on confusing words towards the end, as suggested by the authors. It is, thus, surprising that BERT is able to perform so well on the task, despite this behavior.

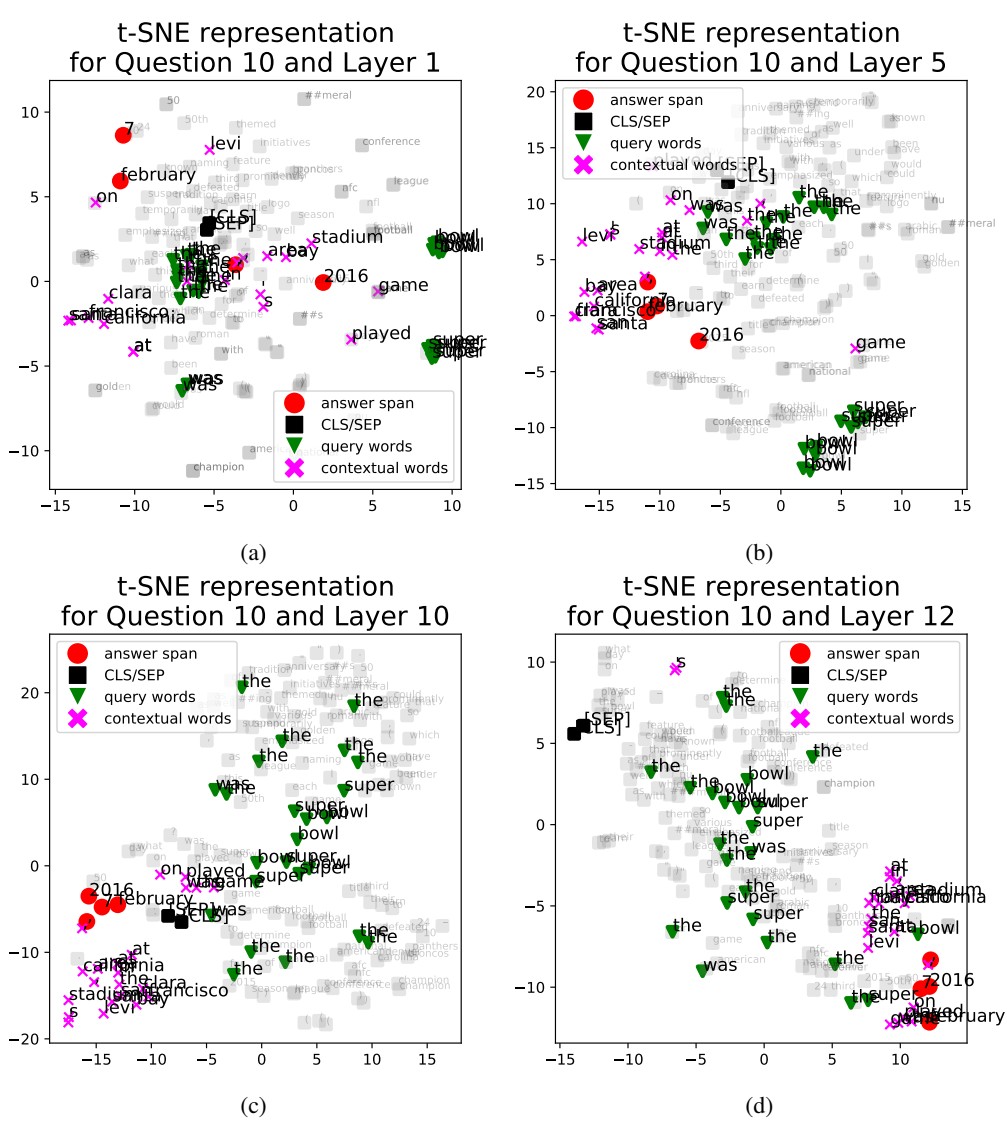

Figure 3: t-SNE Plots for Layers 1, 5, 10, 12 on a SQuAD Example

### 4.1.6 Quantifier Questions

For Quantifier Questions, there are two claims - 6 and 7. For the first claim, we perform experiments mentioned in §3.4.6 to find out the percentages of Numerical Words in top-5 words out of all such words in the passage. We observe that this ratio increases as we go higher up in the layers (SQuAD - L1-6.83%, L11-9.44%, L12-9.93%; DuoRC - L1-36.21%, L11-51.98%, L12-53.04%). The ranges of numbers are different from the original paper, but the suggested trend is followed. A reason for this deviation can be differences in counting of Numerical Words. The author mentioned that they only used cardinal (CD) Parts-of-Speech during our discussion. But, a lot of Quantifier Questions do not have cardinal (CD) words in the corresponding context. The confidence scores for Non-Quantifier Questions (SQuAD - 79.04% ; DuoRC - 86.33%), are significantly lower than for Quantifier Questions with more than one Numerical Word (SQuAD - 85.24% ; DuoRC - 91.20%). Our EM scores on Quantifier Questions are 86.73% for SQuAD , and 54.65% for DuoRC. This shows that even when BERT finds quantifier words increasingly important towards the end layers, it is still able to perform very well on Quantifier Questions. Our results validate claims 6 and 7.

### 4.2 Results beyond the paper

#### 4.2.1 An alternative way of using Integrated Gradients

Before a thorough discussion with the authors, we performed Integrated Gradients in a different manner - we used the ground truth positions as targets for attribution calculation, and the logit outputs (instead of softmax outputs). Note that this would make all the Layer 12 output attributions zero except for two tokens (which have the ground truth start and end positions). When using softmax outputs, the token hidden states affect each other in Layer 12 because of the the normalization term used in softmax, and hence all tokens get some attributions. The categories for the words were also chosen based on the ground truth answers. Multiple features could be sampled from a single answer as the best features per example were not used. Surprisingly, the results observed were similar to the paper and can be referred to in Appendix B. This means that the claims 1 and 2 hold true for both the ways that we calculate IG.

#### 4.2.2 JSD Heatmaps for multiple K values

We plot JSD heatmaps for different values of K (Appendix A). We see that as we increase K, the range of the values on the heatmaps reduce. This means that layers tend to focus on similar words after the first few values of K. For K=5, our top-5 retained heatmap has a range of 0.06-0.60 for SQuAD, and 0.06-0.59 for DuoRC. When K is increased to 10, the top-10 retained heatmap has a range of 0.06-0.56 for SQuAD, and 0.07-0.57 for DuoRC. We expect these ranges to reduce further as we increase the value of K as the words/tokens will get progressively more similar. Thus, limiting K to 5 seems like a good decision on behalf of the authors.

## 5 Discussion

Due to lack of computational resources and time constraints, we were unable to perform IG on all dev samples, and thus chose 1000 random samples per dataset. This can affect the results and statistics significantly. Additionally, the results for DuoRC do not match very well with the authors due to several factors which have been mentioned throughout the report. At the same time, we also show that the authors' claims hold true even for a fraction of the dev set for most cases, which strengthens their claims. Through our code, we also provide a system where researchers can extend this analysis to other datasets by just defining a dataset class similar to ours, specifically a method which converts the dataset into SQuAD format. We also experimented with an alternative way of performing Integrated Gradients described in §4.2.1. The results based on the same align with those of the authors and further strengthen their claim.

### 5.1 What was easy

The authors describe the IG algorithm in their paper, and also provide the link to the code they used to fine-tune BERT. This helped us to prepare the fine-tuning code easily, and find the correct hyperparameters accordingly. Using HF Datasets and Transformers reduced our workload significantly. Also, many popular articles and tutorials exist for fine-tuning BERT on SQuAD for both frameworks PyTorch and TensorFlow, which can be referred to for any help required with implementation. The pair-wise JSD calculation across the layers was also simple with the help of dit [15]. The plotting of heatmaps, qualitative visualization, and t-SNE scatter plots was also easy because of very-well documented libraries - Seaborn and Matplotlib.

### 5.2 What was difficult

Since the authors do not provide the original code at this time of writing, the conversion from DuoRC to SQuAD format was difficult. DuoRC contains examples which have no answers and multiple answers which may or may not exist in the original span. SQuAD, on the other hand has single answers in the training set with a start index provided. This part took a lot of time and computation unnecessarily. IG has also not been described in the paper in great detail, despite the mention of the algorithm. One can use ground truth, max softmax logits, or the predicted positions for target positions. Similarly, the output considered can be the logits or softmax output of the logits which would change the attribution values significantly. This was only clarified through back-and-forth communication with the authors. Finally, the exact details of how the numerical words are counted, what is done when a word is both contextual and a query words, etc. are also not mentioned in the paper, and we had to make our own choices after discussing with the authors.

### 5.3 Communication with original authors

We communicated with the author - Sahana Ramnath - very frequently for over two weeks. We list all the significant questions we asked on our repository added as supplementary material.

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

# Appendix A

## JSD Heatmaps with K=5,10

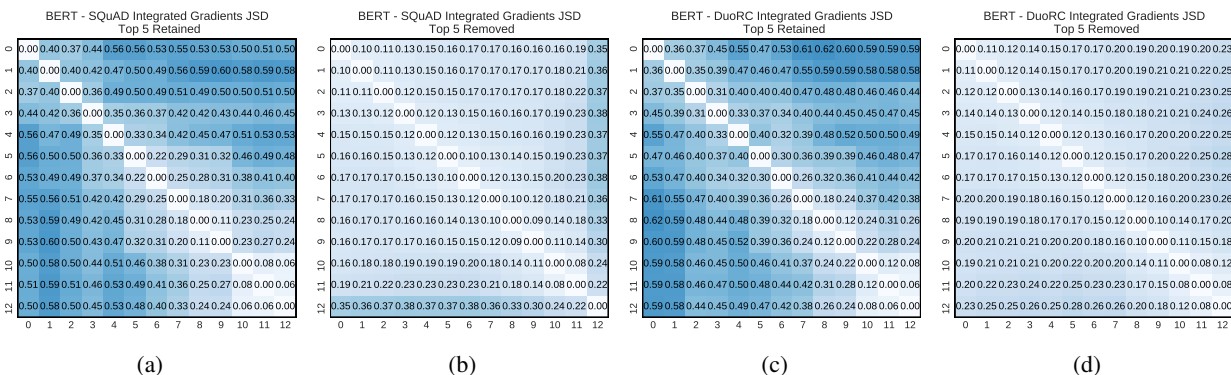

Figure 4: Jensen-Shannon Divergence Heatmaps for K=5

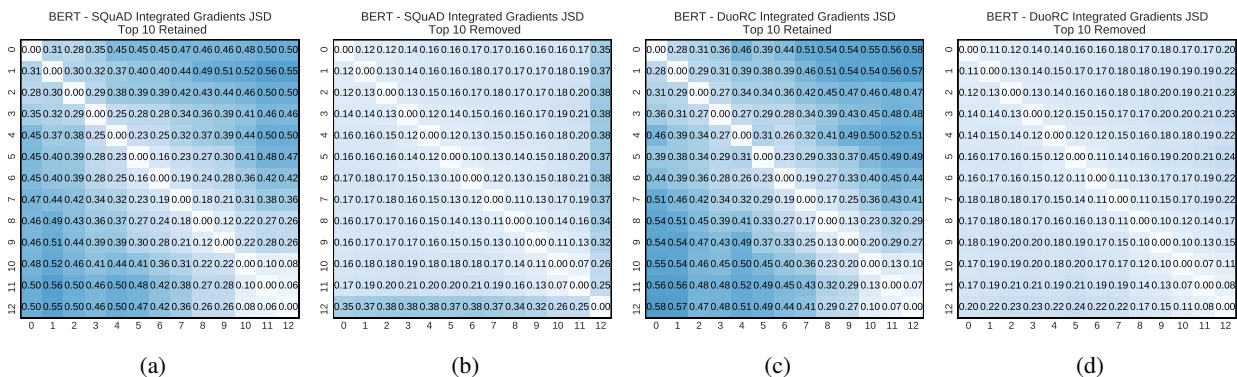

Figure 5: Jensen-Shannon Divergence Heatmaps for K=10

# Appendix B

## Old Integrated Gradients Results

This appendix reports the results we have discussed in §4.2.1:

1. **JSD Heatmaps** - A range of 0.01-0.85 (L1-L12) for SQuAD and 0.03-0.91 (L1-L12) for DuoRC in the case where top-2 scores were retained. For the case where top-2 scores were removed, the ranges were 0.04-0.51 (L1-L12) and 0.04-0.36 (L1-L12) for SQuAD and DuoRC.

2. **QA Functionality** - For SQuAD, we observed similar trends in the percentages of Answer Words(L1- 30.50%, L11 - 38.18%, L12 - 27.84%). The Query Words (24.58%(L1) - 4.62%(L12)) and Contextual Words (32.66%(L1) - 34.24%(L12)) also followed the trends. For DuoRC, however, only the query words (18.66% (L1) - 3.70% (L12) followed a decreasing trend. Contextual (7.36%(L1) - 6.66%(L12))and Answer Words (4.38%(L1) - 3.26%(L12)) in DuoRC remain more or less constant across the layers. Refer Tables 3 and 4.

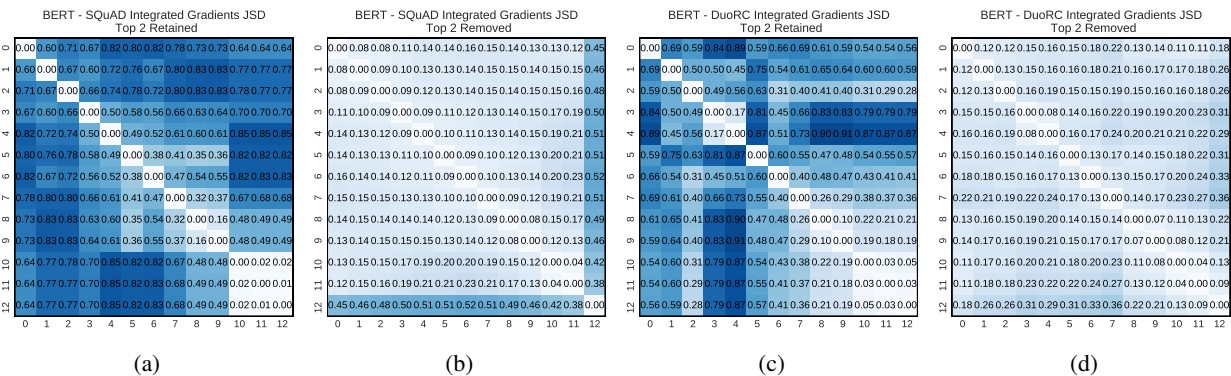

Figure 6: Old Jensen-Shannon Divergence Heatmaps for K=2

| Layer Name | Answer Words% | Contextual Words% | Q-Words% |
|---|---|---|---|
| Embedding | 32.06 | 33.10 | 25.50 |
| Layer 1 | 30.50 | 32.66 | 24.58 |
| Layer 2 | 31.84 | 32.72 | 26.40 |
| Layer 3 | 34.18 | 34.06 | 23.64 |
| Layer 4 | 33.98 | 36.24 | 20.66 |
| Layer 5 | 33.68 | 36.00 | 22.70 |
| Layer 6 | 32.08 | 39.50 | 18.50 |
| Layer 7 | 31.70 | 41.68 | 15.54 |
| Layer 8 | 35.00 | 40.80 | 13.68 |
| Layer 9 | 35.20 | 42.06 | 10.12 |
| Layer 10 | 37.72 | 40.76 | 9.10 |
| Layer 11 | 38.18 | 37.54 | 9.14 |
| Layer 12 | 27.84 | 34.24 | 4.62 |

Table 3: Old Semantic statistics of top-5 words - SQuAD

| Layer Name | Answer Words% | Contextual Words% | Q-Words% |
|---|---|---|---|
| Embedding | 4.40 | 6.88 | 20.72 |
| Layer 1 | 4.38 | 7.36 | 18.66 |
| Layer 2 | 4.50 | 7.16 | 18.00 |
| Layer 3 | 4.70 | 7.62 | 19.36 |
| Layer 4 | 4.54 | 7.52 | 23.10 |
| Layer 5 | 4.58 | 7.22 | 20.08 |
| Layer 6 | 4.44 | 7.48 | 16.12 |
| Layer 7 | 4.34 | 7.02 | 12.80 |
| Layer 8 | 4.88 | 7.50 | 10.58 |
| Layer 9 | 4.50 | 7.00 | 5.80 |
| Layer 10 | 4.32 | 6.70 | 4.48 |
| Layer 11 | 4.40 | 6.80 | 4.06 |
| Layer 12 | 3.26 | 6.66 | 3.70 |

Table 4: Old Semantic statistics of top-5 words - DuoRC

## Appendix C

### PoS Functionality

Based on the paper's Appendix, we also calculate PoS ratios in top-5 words using NLTK's PoS Tagger. Our PoS Tables - Table 5 for SQuAD and Table 6 for DuoRC are shown. We observe that all layers are majorly focused on entity based words with Nouns being 50%-60% of the top-5 words in SQuAD and 60%-70% in DuoRC. However, in comparison with the authors, we observe slightly higher importance to stop words and slightly lower importance to adjectives in the top-5 words. The importance on punctuation marks is also higher.

## Appendix D

### Qualitative Visualization for DuoRC

We visualize an interesting example for DuoRC in Figure 7. DuoRC examples contains long passages, which are broken into several features when we use a maximum length of 384 tokens. Also, since we allow examples to have no answers, we store the first feature (since there is no best feature) in the predictions in case of no answers. In Figure 7, although the original example had no predicted answer, the BERT model is still giving importance to the actual answer *"Chi-Chi"* for the question provided. This means that there was another feature which predicted *"no answer"* with a higher score than this feature which predicted *"Chi-Chi"*. However, there is a good chance that the other feature did not have access to this portion of the context tokens and thus predicted *"no answer"*. This implies that we need to re-think the way we postprocess the predicted answers for each example, as some features might predict correct answers despite having a lower overall score. We will explore this further in the future.

| Layer Name | % nouns | % verbs | % stop words | % adverbs | % adjectives | % punct marks | % words in answer span |
|---|---|---|---|---|---|---|---|
| Embedding | 58.88 | 8.96 | 12.02 | 2.12 | 6.78 | 7.34 | 38.10 |
| Layer 1 | 55.94 | 8.28 | 12.54 | 1.96 | 7.18 | 9.98 | 37.58 |
| Layer 2 | 57.10 | 10.12 | 13.08 | 2.44 | 6.78 | 7.20 | 37.10 |
| Layer 3 | 55.58 | 9.18 | 14.40 | 2.42 | 6.74 | 7.48 | 41.00 |
| Layer 4 | 51.22 | 8.66 | 17.54 | 2.00 | 6.14 | 10.52 | 40.42 |
| Layer 5 | 51.60 | 8.58 | 19.36 | 2.30 | 7.00 | 6.56 | 40.82 |
| Layer 6 | 48.04 | 8.90 | 20.30 | 2.18 | 6.26 | 9.70 | 40.74 |
| Layer 7 | 48.80 | 8.18 | 18.26 | 1.84 | 5.90 | 11.98 | 40.06 |
| Layer 8 | 52.72 | 7.74 | 18.02 | 2.00 | 6.06 | 7.78 | 41.90 |
| Layer 9 | 50.42 | 6.24 | 17.44 | 1.96 | 5.98 | 11.74 | 41.18 |
| Layer 10 | 53.00 | 5.98 | 18.20 | 1.88 | 5.90 | 10.08 | 43.36 |
| Layer 11 | 57.90 | 3.78 | 15.06 | 1.76 | 6.30 | 8.26 | 42.52 |
| Layer 12 | 54.86 | 3.68 | 15.14 | 1.74 | 5.90 | 12.12 | 42.94 |

Table 5: PoS statistics of top-5 words - SQuAD

| Layer Name | % nouns | % verbs | % stop words | % adverbs | % adjectives | % punct marks | % words in answer span |
|---|---|---|---|---|---|---|---|
| Embedding | 72.72 | 7.40 | 8.26 | 1.36 | 4.02 | 6.28 | 11.78 |
| Layer 1 | 66.56 | 7.48 | 11.62 | 1.32 | 3.68 | 9.54 | 11.70 |
| Layer 2 | 67.16 | 6.64 | 9.12 | 1.34 | 3.68 | 11.58 | 12.60 |
| Layer 3 | 67.20 | 6.92 | 11.70 | 1.46 | 3.98 | 8.86 | 13.36 |
| Layer 4 | 68.38 | 7.04 | 13.34 | 1.34 | 3.56 | 6.74 | 13.16 |
| Layer 5 | 67.52 | 7.16 | 14.08 | 1.44 | 3.72 | 6.36 | 12.68 |
| Layer 6 | 62.64 | 8.28 | 15.48 | 1.08 | 3.26 | 10.40 | 12.96 |
| Layer 7 | 61.24 | 6.30 | 14.26 | 1.12 | 2.86 | 15.02 | 12.68 |
| Layer 8 | 69.86 | 4.14 | 10.78 | 1.24 | 3.56 | 9.58 | 13.36 |
| Layer 9 | 74.72 | 2.34 | 6.40 | 1.18 | 2.72 | 11.60 | 12.66 |
| Layer 10 | 69.66 | 2.58 | 8.04 | 1.18 | 3.26 | 14.88 | 12.90 |
| Layer 11 | 66.38 | 2.38 | 10.10 | 1.24 | 3.66 | 15.74 | 13.06 |
| Layer 12 | 72.08 | 2.80 | 11.10 | 1.36 | 3.86 | 8.36 | 12.94 |

Table 6: PoS statistics of top-5 words - DuoRC

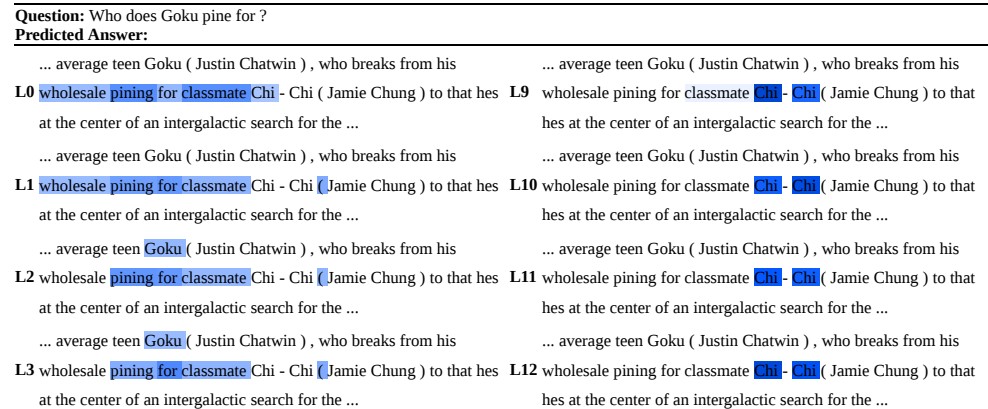

Figure 7: Qualitative Visualization of top-5 words - DuoRC

# Appendix E

## Recommendations to the authors

While we understand that the authors had to adhere to a certain page limit, the following information, if added, maybe as a supplementary material with the paper, could prove to be beneficial for the reproducibility of the paper:

- **The settings for Integrated Gradients**: Specifically, the choice of target positions, how are start and end attributions combined, what kind of target outputs (softmax/logits) are chosen, and which features for each example are chosen, can be mentioned in detail in order to make the results reproducible. Additionally, whether the Jensen-Shannon Divergence is calculated on words or tokens can be clearly specified. Similarly, for the rest of the analysis, whether it is performed on the token-wise importance scores or word-wise importance scores can be clarified.

- **The pre-processing and post-processing details for DuoRC SelfRC dataset**: SQuAD being a simpler dataset does not usually cause issues, but training a dataset like DuoRC which has combination of abstractive

and extractive question-answering tasks using a span-prediction model is relatively complex. Which examples in which stage are discarded and how is the answer chosen in each of the cases mentioned in 3.2.2 can be mentioned. During the post-processing, whether or not the question is allowed to have "no answer" as output can also be added.

- **Quantifier Questions**: The information for finding the quantifier questions can be added. For example, we had to search for 'how man' instead of 'how many' to get all such questions. This detail was not mentioned in the paper. Additionally, how are the Numerical Word percentages calculated can also be described. This detail is very important to reproduce the tables with high precision. There can be multiple ways of counting Numerical Words depending on what kind of words are defined as Numerical Words. How the search is performed, how is the data tagged, what kind of Part-of-Speech tagger is used, etc. could also be added. The equation used for confidence calculation could also be shown because it can be calculated in several ways.

- **Categorizing the words for t-SNE representation**: There may be words which belong to both *query words* and *contextual words*. The authors can describe how this conflict is resolved. Additionally, how are *contexual words* chosen, and whether ground truth or predicted answer tokens are used for categorization can also be mentioned.

- **Alternatives to t-SNE**: We note that t-SNE representation varies significantly with random initialization, and there could be better, relatively stable choices for dimensionality reduction. Techniques like Principal Component Analysis (PCA), Uniform Manifold Approximation and Projection (UMAP) can also be used as alternatives. This will make the visualizations more reproducible. Alternatively, a short description of why t-SNE was chosen can also be added.

# Appendix F

**Ethical Considerations**

Since Machine Learning systems will compound and propagate the personal biases that are incorporated when humans prepare datasets, it is important to mention the possible causes and biases that will be present in our training datasets — SQuAD and DuoRC here - so that skewness is minimized and fairness is maximized while interpreting/looking at results of our model. We also mention that the impact of fairness in Reading Comprehension tasks, in general, is not as severe as in other tasks that directly impact business/important decisions like job recommendations [20], face-based datasets [21] etc. At most, it will lead to a reader's demographic preference for questions from the corpus.

- The SQuAD dataset consists of questions posed by crowdworkers on a set of Wikipedia articles, the demographic distribution and composition, qualification/education, their environment of the crowdworkers will affect the kind of questions they pose on the Wikipedia articles. The people who have authored those articles would have written it from their own perspective. The relationship between the author-reader perspectives will affect the kind of questions-answers that are prepared.

- The DuoRC dataset consists of QA pairs from two different versions of 7680 plots from IMDb and Wikipedia. The racial, gender bias incorporated in those movies, in addition to those of the reviewers, plot authors,etc. would be incorporated in the QA pairs. Additionally, any sarcasm, jokes present in the movie plots will be given equal importance as any other dialogue/plot, which may result in unfair or biased results during the prediction.

However, the fairness of our model is of a lower-priority in this analysis since there will not be any disparate impact of our findings on any minority/historically disadvantaged group. We do not use sensitive attributes in our study. Our focus, in line with that of the authors of the paper, is more on addressing answerability, accuracy, interpretability of BERT models. This means that the approach presented will work efficiently with either biased/unbiased data.

Since the BERT model is itself pre-trained on a Wikipedia-based corpus, it is very much possible that unethical statements/bias are ingrained in its parameters in some form. Indeed, there has been work on finding and mitigating social/intersectional/gender biases in contextualized word representations present in BERT [22–24]. Having mentioned that, we believe that fine-tuning on a large enough dataset with unbiased examples could mitigate the issue to some extent. Reiterating, the approach presented in the original paper and discussed here is purely mathematical in nature, and will work equally well on any dataset, biased or unbiased, given enough number of examples.

