# OpenReview forum: "Reproducibility Report: Towards Interpreting BERT for Reading Comprehension Based QA"
_ML_Reproducibility_Challenge/2020 — Reject_

### Official Review · AnonReviewer3 · 2021-02-25
**Review of Towards Interpreting BERT for Reading Comprehension QA**

**Rating:** 9
**Confidence:** 2

**Review:**


- Reproducibility Summary: the paper includes the reproducibility summary on the first page. The summary is clear, well written, and major findings (i.e. successful replication within a good margin) are incorporated in the summary.

- Scope of reproducibility: it is well described: specifically the authors investigate the roles of different BERT layers in the context of Reading Comprehension based Question Answering (RCQA). The authors list all the reproduced/verified claims from the original paper.

- Code: the authors re-implement the original approach since the official code is not available. The code folder contains a well-documented readme file and requirements are specified.

- Communication with original authors: the authors had email interactions with the original authors. The questions and answers are included in the supplementary material.

- Hyperparameter Search: the original authors did not provide any code, thus the authors could not re-use it for hyperparameter search. The authors report the selected hyperparameters, but they do not mention whether they did a grid search on the hyperparameter space.

- Ablation Study: there is no ablation study, but it does not make sense to have it.

- Discussion on results: the authors obtain F1 scores that are comparable to those in the original paper, but their model slightly underperforms with respect to the original one. The authors do not mention why the results are different and what are the plausible causes. Apart from this, the experimental results are well presented and compared with the original results. The authors describe easy parts and challenges in reproducing the original paper.

- Recommendations for reproducibility: the authors do not explicitly mention any recommendations for the original authors, but they do describe missing details that were important for the reproducibility.

- Results beyond the paper: the authors present some results beyond the original paper: they implemented Integrated Gradients (IG) in a different way and extended the original analysis of Jensen-Shannon Divergence heatmaps with different cut-offs.

- Overall organization and clarity: the paper is clear and well organized. I list some typos in the following:
    - Line 45: t-SNE -> please expand the acronym at least when first mentioned;
    - Line 81: don’t -> do not
    - Line 149: we represent show

Overall evaluation:
Pros:
- The paper is clear and well written;
- The authors could reproduce (within a decent margin) the results of the original paper;
- The authors performed some extra analyses beyond the original paper.

Cons:
- Some experimental results could have been explained with more details.


**Familiar With The Original Paper:**

I have not read the original paper

**Reproducibility Summary:**

Report has summary

---

### Official Review · AnonReviewer2 · 2021-02-25
**Reproduce experiment with BERT for reading comprehension QA**

**Rating:** 9
**Confidence:** 3

**Review:**

The reproduced work of "towards interpreting BERT for reading comprehension QA" is well organized and clearly explained. The authors rebuilt the experiment from scratch due to the availability of the original paper code. The implementation and scope align with the original paper. A large part of the results matched, and the difference was explained which was mostly due to computation limit and sample difference. Overall, I'd recommend accepting.

**Familiar With The Original Paper:**

I have read the original paper

**Reproducibility Summary:**

Report has summary

---

### Official Review · AnonReviewer1 · 2021-03-04
**Reproducibility Report: Towards Interpreting BERT for Reading Comprehension QA**

**Rating:** 6
**Confidence:** 4

**Review:**

Thank you for this carefully written paper that focused on  interpreting BERT for reading comprehension questions and answers (Q&A). It was successful in reproducing the main findings of its chosen paper, without an originally released codebase to build on. Obtaining this outcome, required, as expected regular interaction with the first author of the chosen paper. My major comments or suggested improvements relate to documenting the  reading comprehension Q&A; the authors excelled in reporting on the solution (processing methods, evaluation methods, processing outcomes), and even had findings beyond the chosen paper. However, the problem of  reading comprehension Q&A, its relevant literature, and implications to reading comprehension QA practice were not addressed. More over, I would have expected to see a statement relating human subject ethics in the paper, although its data originated from a previously openly released corpus. This contradiction between the (currently weak) domain substance and (currently outstanding) computing contributions made me rate the paper as Marginally above acceptance threshold.

**Familiar With The Original Paper:**

I have read the original paper

**Reproducibility Summary:**

Report has summary

---

### Decision · Program_Chairs · 2021-03-31

**Decision:**

Reject

**Comment:**

Overall reviews and/or the paper content not good enough for the AC to recommend to the journal.